# SIMILE: Introducing Sequential Information towards More Effective Imitation Learning

## Abstract

Reinforcement learning (RL) is a metaheuristic aiming at teaching an agent to interact with an environment and maximizing the reward in a complex task. RL algorithms often encounter the difficulty in defining a reward function in a sparse solution space. Imitation learning (IL) deals with this issue by providing a few expert demonstrations, and then either mimicking the expert's behavior (behavioral cloning, BC) or recovering the reward function by assuming the optimality of the expert (inverse reinforcement learning, IRL). Conventional IL approaches formulate the agent policy by mapping one single state to a distribution over actions, which did not consider sequential information. This strategy can be less accurate especially in IL, a weakly supervised learning environment, especially when the number of expert demonstrations is limited.

This paper presents an effective approach named Sequential IMItation LEarning (**SIMILE**). The core idea is to introduce sequential information, so that an agent can refer to both the current state and past state-action pairs to make a decision. We formulate our approach into a recurrent model, and instantiate it using LSTM so as to fuse both long-term and short-term information. SIMILE is a generalized IL framework which is easily applied to BL and IRL, two major types of IL algorithms. Experiments are performed on several robot controlling tasks in OpenAI Gym. SIMILE not only achieves performance gain over the baseline approaches, but also enjoys the benefit of faster convergence and better stability of testing performance. These advantages verify a higher learning efficiency of SIMILE, and implies its potential applications in real-world scenarios, *i.e.*, when the agent-environment interaction is more difficult and/or expensive.

## 1 Introduction

In recent years, Reinforcement learning (RL) (Sutton & Barto, 1998) has been widely used in many research areas. Different from supervised learning (SL) in which a set of input/output pairs are provided, the fundamental idea of RL is to interact with an environment and learn from trial-and-error. The goal of an RL algorithm is to find a policy that maximizes the total reward in a complete episode. However, as the learning task becomes more and more complex, researchers often encounter the difficulty in defining a reasonable reward function. To alleviate this issue, imitation learning (IL) was proposed (Schaal, 1999), which instead provides a few expert demonstrations for the agent to learn from. Based on the assumption that the expert is always the best, there are two types of IL approaches, known as behavioral cloning (BC) (Abbeel & Ng, 2004) and inverse reinforcement learning (IRL) (Ng & Russell, 2000), respectively[1]. Both of them train the agent so that its behavior is indistinguishable from that of the expert – BC tries to mimic the behavior of the expert, and IRL aims at recovering the underlying reward function so that its inner-loop, an RL algorithm, can learn from it.

A noticeable challenge of IL lies in the limited amount of expert demonstrations, mainly due to the potential time and financial costs especially in real-world environments. On the other hand, conven-

---

[1] Throughout this paper, we use "imitation learning" (IL) to name all the RL methods which learn from expert demonstrations. Among them, all methods which explicitly recover the reward function are named "inverse reinforcement learning" (IRL), otherwise "behavioral cloning" (BC). For example, GAIL (Ho & Ermon, 2016) belongs to BC because it bypassed the reward function and directly generate the agent policy.

tional IL approaches often interpret the behavior of an expert using individual state-action pairs. We argue that this introduces ambiguity and noise into the training process, as one single state may be insufficient to provide all necessary information for the agent to make a decision. For example, in playing a video game (Bellemare et al., 2013), each state contains a still image, which does not tell the moving direction and speed of each object. In such cases, the expert can take completely different actions under the "same" situation (two states seem identical in the defined representation, but the expert can refer to extra information and realize the difference), which confuses the agent and introduces additional noise to the training process. Please refer to Section 3.2 for detailed examples.

This paper provides a straightforward approach named Sequential IMItation LEerning (**SIMILE**), which allows the agent to see sequential information (*i.e.*, past state-action pairs) in order to acquire extra information – as an intuitive example, in playing a video game, the moving direction and speed of an object can be predicted by seeing a few consecutive states. Therefore, we formulate the policy into a recurrent model, and instantiate it using an LSTM (Hochreiter & Schmidhuber, 1997) which effectively fuses long-term and short-term information. SIMILE is a generalized algorithm which can be instantiated in two important examples of IL, namely IRL and BC. In either case, we also fuse the information produced by each LSTM cell using AdaBoost (Freund & Schapire, 1997), a sequential classifier which adjusts the weight of each state according to its importance.

We evaluate SIMILE in OpenAI Gym (Brockman et al., 2016), a popular benchmarking environment for RL algorithms. We investigate eight robot controlling tasks, varying from basic objects to complex motion systems. Beyond both IRL and BC baselines, our approach achieves consistent overall performance gain – in some tasks, the trained agent performs comparably or even better than the expert. SIMILE also enjoys two additional benefits. First, the training process converges faster, which not only remedies the drawback in running speed, but also reduces the number of interaction with the environment which can be expensive. Second, the testing performance becomes more stable in terms of the standard deviation of total reward, suggesting that the agent becomes more reliable. Both these advantages imply that our approach makes use of training data more effectively, and owe to the exploitation of sequential information. Moreover, these advantages also suggest that our approach is more friendly to training an RL system in real-world environments.

The remainder of this paper is organized as follows. Section 2 briefly reviews related work. Section 3 introduces the SIMILE approach and its application to IRL and BC pipelines. After experiments are shown in Section 4, we conclude this work in Section 5.

## 2 RELATED WORK

Reinforcement learning (RL) (Kaelbling et al., 1996; Sutton & Barto, 1998) is an area of machine learning concerned with how software agents ought to take actions in an environment so as to maximize some notion of cumulative reward[2]. It implies, and has made significant progresses in a wide range of applications, including robot navigation (Zhu et al., 2017), urban route planning (Kuderer et al., 2015), training multi-agent systems (Mnih et al., 2015), playing video games (Bellemare et al., 2013) and chess games (Silver et al., 2016), *etc*. Compared to supervised learning (SL) which provides a set of input-output pairs, RL often trains an agent by facilitating it to interact with an environment and get trained from trial-and-error. An RL system is often formulated by a Markov Decision Process (MDP) in which a reward function is defined at each state transition, and the goal is to maximize the total reward in a pre-defined task, *i.e.*, a complete episode. One of the key issues in RL is to sample effective training data from the environment, for which a few popular algorithms were proposed, such as Monte-Carlo sampling (Sutton & Barto, 1998; Dosovitskiy & Koltun, 2017), temporal differencing (Sutton, 1988; Amiranashvili et al., 2018) and deep Q-learning (Mnih et al., 2015; Wang et al., 2015). To improve training efficiency, off-policy or asynchronous methods were also designed (Schulman et al., 2015; Mnih et al., 2016; Schulman et al., 2017) for parallel optimization. The strategy of combining different sampling tricks achieves superior performance was studied in (Hessel et al., 2017).

One of the key factors of RL is to define a reward function, however, this is difficult in many complex scenes. Imitation learning (IL) was then proposed to deal with this problem. The core idea of IL is to provide a few expert demonstrations, and ask the agent to mimic the behavior of the expert.

---

[2] https://en.wikipedia.org/wiki/Reinforcement_learning

There are mainly two types of methods in IL. The first type is named behavioral cloning (BC) or apprenticeship learning (Abbeel & Ng, 2004; Abbeel et al., 2010; Hadfield-Menell et al., 2016), in which a set of state-action pairs was extracted from the expert demonstrations, and the agent learned from these data like the supervised setting. This method can be generalized to optimizing an agent policy that is indistinguishable from the expert policy (Ho et al., 2016), and thus be formulated in the form of a generative adversarial network (GAN) (Ho & Ermon, 2016). The second type called inverse reinforcement learning (IRL) (Ng & Russell, 2000) instead aimed at recovering the underlying reward function from expert demonstrations, so that a standard RL algorithm can learn from the reward function. Therefore, IRL is often equipped with an inner-loop which is an RL algorithm, and the training process alternatively updates the agent policy and the reward function, assuming that the expert always makes the optimal choice. Previous studies have included max-margin or max-entropy approaches (Ziebart et al., 2008; Wulfmeier et al., 2015) as well as probabilistic approaches (Ramachandran & Amir, 2007; Levine et al., 2011) into IRL. The equivalence of IRL and GAN was later proved in (Finn et al., 2016), which leads to adversarially training an IRL system towards stronger transfer ability (Fu et al., 2018).

This work aims at introducing sequential information into both IRL and BC frameworks. This recalls the family of recurrent neural networks (RNNs) (Hopfield, 1982; Graves et al., 2009), and a popular instantiation of RNNs named long short-term memory (LSTM) (Hochreiter & Schmidhuber, 1997) which has shown its ability of incorporating sequential information in many problems such as natural language processing (Kumar et al., 2016), video processing (Donahue et al., 2015) and reinforcement learning (Parisotto & Salakhutdinov, 2018). Also, the sequential outputs from LSTM need to be combined into a discriminator, which is related to the problem of fusing several weak classifiers into a strong one. This problem was investigated in Adaptive Boosting (AdaBoost) (Freund & Schapire, 1997) and its variants such as the multi-class AdaBoost (Hastie et al., 2009).

## 3 APPROACH

### 3.1 BACKGROUND

The environment of an RL system can be defined as a quintuple $\langle \mathcal{S}, \mathcal{A}, P, \gamma, R \rangle$ abstracting a finite-state Markov Decision Process (MDP). $\mathcal{S}$ denotes the space of all possible states, and $\mathcal{A}$ denotes the set of actions. Given a pair of state $s \in \mathcal{S}$ and action $a \in \mathcal{A}$, $P(\cdot \mid s, a)$ is the state transition function, *i.e.*, $P(s' \mid s, a)$ is the probability that another state $s'$ is achieved when the agent takes action $a$ at state $s$. As a special case, $P_0(s_0) \doteq P_0(s_0 \mid \varnothing)$ determines the probability of the first state $s_0$. $\gamma \in [0, 1]$ is named discounted factor measuring how much the action of the current state impacts a future reward. $R : \mathcal{S} \times \mathcal{A} \mapsto \mathbb{R}$ denotes the reward function to measure the benefit of a state-action pair. $R$ is often bounded by $R_{\max}$ in its absolute value: $|R(\cdot, \cdot)| \leqslant R_{\max}$.

The goal of RL is to learn a *policy*, *i.e.*, a function $\pi : \mathcal{S} \times \mathcal{A} \mapsto [0, 1]$ which defines the probability of each action $a$ to be taken by the agent under state $s$. Given a policy $\pi$ and a starting state $s_0$, the MDP can sample an action at each time $t \in \mathbb{N}$ and then go to the next random state:

$$a_t \sim \pi(s_t, \cdot), \qquad s_{t+1} \sim P(\cdot \mid s_t, a_t). \tag{1}$$

This produces a trajectory $\tau = \{s_0, a_0, \ldots, s_{T-1}, a_{T-1}, s_T\}$, and we denote it by $\tau \sim \mathcal{P}(\pi, P \mid s_0)$. Compared to directly learn policy $\pi$, it is often easier to learn from a *critic*, *e.g.*, a function that implicitly defines the optimal policy $\pi^\star$. An example is to construct a value function $V_\pi(\cdot)$ which estimates the reward obtained by policy $\pi$ at each state $s$:

$$V_\pi(s) = \mathbb{E}\left[\sum_{t=0}^{\infty} \gamma^t \times R(s_t, a_t \mid s_0 = s)\right], \tag{2}$$

and the optimal policy $\pi^\star$ satisfies $V_{\pi^\star}(s) \geqslant V_\pi(s)$ at each state $s$ and all policy $\pi$. Another critic is named $Q$-function, which estimates the reward obtained by policy $\pi$ at each state $s$ if action $a$ is taken as the first step (and all the remaining steps are determined by $\pi$):

$$Q_\pi(s, a) = \mathbb{E}\left[\sum_{t=0}^{\infty} \gamma^t \times R(s_t, a_t \mid s_0 = s, a_0 = a)\right]. \tag{3}$$

Once a $Q$-function is constructed on a policy $\pi$, we can continue to obtain a better policy $\pi'$ (namely, $V_\pi(s) \leqslant V_{\pi'}(s)$ for all states $s \in \mathcal{S}$) by setting $\pi'(s, \arg\max_a\{Q_\pi(s, a)\}) = 1$ at each state $s$.

**Acrobot**: a robot controlling task | **SpaceInvader**: a video game of Atari

Input state: two angles between link and horizon, the velocity of two ending points
Hidden information: the mass of two links and gravity, therefore the acceleration of two ending points

Input state: image pixels
Hidden information: the moving direction and speed of enemies, the speed of bullets, the maximum speed of the player, the remaining volume of the blindages

Figure 1: Two examples that the state vector is insufficient for decision making. The left figure show Acrobot (Sutton, 1996), and the right figure show an Atari game named SpaceInvaders (Bellemare et al., 2013) which takes image data as input. See texts for detailed descriptions.

In many scenarios, the reward function $R$ is difficult to define. IL starts with constructing a set of expert demonstrations $\{\tau_1, \tau_2, \ldots, \tau_K\}$. Each $\tau_k$ is named a trajectory which records the expert's behavior in a complete episode $\tau_k = (s_{k,0}, a_{k,0}, s_{k,1}, a_{k,1}, \ldots, s_{k,T_k})$. This is to implicitly define an expert policy $\pi_E$. Then, based on the assumption that the expert always performs the best, there are two ways to train the agent. BC uses all pairs $(s_{k,t}, a_{k,t})$ as training data to optimize the agent policy $\pi$ in a supervised manner, and IRL updates $\pi$ and the reward function $R$ alternatively so that $R$ always prefers $\pi_E$ to $\pi$. Mathematically, both BC and IRL solves the following minimax problem:

$$R^\star = \arg\min_R \left\{ \max_\pi \left\{ \lambda \times H(\pi) + \mathbb{E}_\pi[R(s_0 \sim P_0)] \right\} - \mathbb{E}_{\pi_E}[R(s_0 \sim P_0)] \right\}, \qquad (4)$$

where $H(\pi)$ is the casual entropy term of $\pi$, measuring how confident the policy $\pi$ is (the larger $H(\pi)$, the more confident, the better (Bloem & Bambos, 2014)). $R$ takes different forms in BC and IRL. In BC, it can be written as $\mathbb{E}_\pi[R(s_0)] = \sum_{t=0}^\infty -\ln D(s_t, a_t)$, where all $(s_t, a_t)$ pairs are sampled from $\pi$ using Equation 1, and $D(s, a)$ is a discriminative function to judge the likelihood that $(s, a)$ comes from the decision of an expert. In IRL, $\mathbb{E}_\pi[R(s_0)] = V_\pi(s_0)$ is the value function estimating the total reward obtained by $\pi$. Optimizing Equation 4 is equivalent to GAN (Goodfellow et al., 2014), in which the agent (RL inner-loop) plays the role of a generator, and the reward function (IRL outer-loop) plays the role of a discriminator (Finn et al., 2016). The final goal is to make it difficult for the reward function to distinguish $\pi$ from $\pi_E$.

## 3.2 Motivation: Introducing Sequential Information

Current IL approaches heavily rely on high-quality expert demonstrations. For example, GAIL (Ho & Ermon, 2016) constructed a number of expert demonstrations for each task by running an RL algorithm named TRPO (Schulman et al., 2015) on the ground-truth reward function, and showed that the number of demonstrations does impact IL performance. This process fits well to the virtual environment (*e.g.*, robot controlling games in OpenAI Gym (Brockman et al., 2016)), but in real-world RL problems, it may be intractable (the ground-truth reward does not exist) or very expensive (it may break physical device, or take a very long time). We will show in Section 4.3 that our approach enjoys a larger advantage in the scenarios of fewer expert demonstrations.

We argue that one of the reasons of requiring a large amount of data lies in the exploitation of training data. Recall that the policy $\pi : \mathcal{S} \times \mathcal{A} \mapsto [0, 1]$ decides the next action based on merely one (current) state $s$. This may be insufficient for decision making. Figure 1 shows two examples. The first one is a robot controlling task named Acrobot (Sutton, 1996), in which the agent is asked to actuate the joint of a two-link pendulum to swing it as high as possible. In this case, the state vector contains two angles and the velocity of two ending points, but it does not provide the acceleration of both ending points. Although the acceleration can be determined by physical laws, we note that the agent is often not equipped with this type of knowledge, nor does it know the mass of these two links. Therefore, when an agent is trained in the standard environment (the mass is 1 for both links) and then transferred to another setting (*e.g.*, the mass of the second link becomes 2), it may need to take a different action at the same state. The second example is more intuitive: in an Atari game named SpaceInvaders (Bellemare et al., 2013), the agent is asked to control a spaceship to eliminate as many enemies as possible. In this case, the state variable is the real-time screenshot, in which all objects are still and the agent does not know if the enemies will move towards left or

right, the speed of the enemies, the speed of itself, *etc*. In more complex scenes, there may even exist some factors which helps decision making but are difficult to formulate explicitly in the state variable. We name them as *hidden information*. An expert such as a human or an RL algorithm seeing the ground-truth reward function may take completely different actions at the "same" state with different hidden information. However, an IL algorithm is not aware of these information and thus may be confused by noisy training data (the same input is paired with different outputs). With a limited amount of training data (expert demonstrations), this issue may lead to instability in testing performance (*e.g.*, a large variance), as shown in Section 4.2.

To take hidden information into consideration, a straightforward idea is to introduce sequential information. This is to say, when the agent needs to take an action, it receives all previous state-action records rather than merely one single (current) state as input data. Intuitively, this is partly how humans apperceive the actual state, especially the factors that are not formulated in the state variable. For example, in a video game, the motion properties of all objects become predictable if a few consecutive frames are taken as input.

### 3.3 SIMILE: FRAMEWORK AND TWO EXAMPLES

Therefore, we desire an algorithm to extract information from sequential inputs. Mathematically, we start with Equation 1 and replace the original policy $\pi$ with $\tilde{\pi}$, and sampling strategy $a_t \sim \pi(s_t, \cdot)$ with $a_t \sim \tilde{\pi}(s_t, \cdot \mid s_0, a_0, \ldots, s_{t-1}, a_{t-1})$, *i.e.*, the desired policy $\tilde{\pi}$ determines the distribution of the current action $a_t$ based on the current state $s_t$ and all previous state-action pairs $(s_0, a_0), \ldots, (s_{t-1}, a_{t-1})$.

This is a sequential inference problem which can be formulated using a recurrent model:

$$
\begin{aligned}
a_t \quad &\sim \quad \tilde{\pi}(s_t, \cdot \mid s_0, a_0, \ldots, s_{t-1}, a_{t-1}) = \pi(s_t, \cdot \mid a_{t-1} \sim \tilde{\pi}(s_{t-1}, \cdot \mid s_0, a_0, \ldots, s_{t-2}, a_{t-2})) \\
&= \quad \pi(s_t, \cdot \mid a_{t-1} \sim \pi(s_{t-1}, \cdot \mid a_{t-2} \sim \pi(\ldots \mid a_0 \sim \pi(s_0, \cdot) \ldots))).
\end{aligned} \tag{5}
$$

Intuitively, in the decision process, we need to consider both long-term memory (*e.g.*, if a similar situation appears previously, the agent may refer to the record to predict the possible transitions) and short-term memory (*e.g.*, by seeing a few previous states, it is possible to extract hidden information such as motion properties), so we make use of LSTM (Hochreiter & Schmidhuber, 1997) to fuse these information. LSTM is composed of a series of cells, each of which receives three inputs: a long-term memory signal $\mathbf{h}_t$, a short-term memory signal $\mathbf{c}_t$ and a state variable $s_t$, and outputs a distribution on all possible actions meanwhile updating $\mathbf{h}_t$ and $\mathbf{c}_t$ for the next cell:

$$
(s_{t+1}, \mathbf{c}_{t+1}, \mathbf{h}_{t+1}) = \text{LSTM}(s_t, \mathbf{c}_t, \mathbf{h}_t) \tag{6}
$$

In order to learn the policy, we explicitly decompose this function into two stages:

$$
(a_t, \mathbf{c}_{t+1}, \mathbf{h}_{t+1}) \sim \tilde{\pi}(s_t, \cdot \mid \mathbf{c}_t, \mathbf{h}_t), \qquad s_{t+1} \sim P(\cdot \mid s_t, a_t). \tag{7}
$$

Here we simply apply the vanilla LSTM (Greff et al., 2017). It is possible that an advanced model can lead to performance gain, but this goes out of the discussion scope of our paper. Substitut-

---

**Algorithm 1** Applying SIMILE to vanilla IRL

1: **Input:** expert trajectories $\mathcal{T} = \{\hat{\tau}_1, \hat{\tau}_2, \ldots, \hat{\tau}_K\} \sim \mathcal{P}(\pi_E, P)$, learning rates $\eta_1$ and $\eta_2$;
2: Initialize: policy $\tilde{\pi}$ parameterized by $\boldsymbol{\theta}_0$, reward function $R$ initialized by $\boldsymbol{\omega}_0$;
3: **for** $m = 1, 2, \ldots, M$ **do**
4: $\quad s_{m,0} \sim P_0(\cdot), T_m \leftarrow 0$;
5: $\quad$ **while** task is not accomplished **do**
6: $\quad\quad a_{m,t} \sim \tilde{\pi}(s_{m,t} \mid s_{m,0}, a_{m,0}, \ldots, s_{m,t-1}, a_{m,t-1})$;
7: $\quad\quad s_{m,t+1} \sim P(\cdot \mid s_{m,t}, a_{m,t}), T_m \leftarrow T_m + 1$;
8: $\quad$ **end while**
9: $\quad \tau_m \leftarrow (s_{m,0}, a_{m,0}, \ldots, s_{m,T_m-1}, a_{m,T_m-1}, s_{m,T_m})$;
10: $\quad \boldsymbol{\omega}_{m+1} \leftarrow \boldsymbol{\omega}_m - \eta_1 \times \left\{ \nabla_{\boldsymbol{\omega}_m} \left[ \sum_{t=0}^{T_m-1} R(s_{m,t}, a_{m,t}) - \mathbb{E}_{\hat{\tau}_k \in \mathcal{T}} \left[ \sum_{t=0}^{\hat{T}_k-1} R(\tilde{s}_{k,t}, \tilde{a}_{k,t}) \right] \right] \right\}$;
11: $\quad$ Update $\boldsymbol{\theta}_m$ as $\boldsymbol{\theta}_{m+1}$ using $R$ and an RL algorithm (*e.g.*, TRPO (Schulman et al., 2015));
12: **end for**
13: **Output:** the learned policy $\tilde{\pi}$ parameterized by $\boldsymbol{\theta}_M$.

---

ing Equation 7 into Equation 4 obtains the Sequential IMItation LEarning (SIMILE) algorithm. In the following parts, we will instantiate SIMILE in two types of IL algorithms, namely IRL and BC.

- **Instantiation in Inverse Reinforcement Learning**

First, we apply SIMILE to the vanilla inverse reinforcement learning, in which the goal is to recover the reward function $R$ for an RL algorithm (the inner-loop) to learn from. We initialize $R$ and policy $\tilde{\pi}$ using random noise, and update their parameters $\boldsymbol{\omega}_t$ and $\boldsymbol{\theta}_t$ alternatively. In each iteration, both the expert and agent sample a trajectory using Equation 7. Then, $R$ is updated to maximize the advantage of $\pi_\mathrm{E}$ over $\tilde{\pi}$, and $\tilde{\pi}$ is updated using an RL algorithm in the inner-loop (here we use the TRPO (Schulman et al., 2015) algorithm) which takes $R$ as input.

The pipeline of IRL with SIMILE incorporated is shown in Algorithm 1. Here, the total reward obtained by the agent, $\mathbb{E}_\pi[s_0 \sim P_0]$, is computed by sampling a complete trajectory and summing up all values of the estimated reward function. On the other hand, the total reward obtained by the expert, $\mathbb{E}_{\pi_\mathrm{E}}$, is directly computed by averaging the performance of all experts, *i.e.*, $\mathbb{E}_{\pi_\mathrm{E}}[\tau \sim \mathcal{T}]$ where $\mathcal{T}$ is the set of expert trajectories.

- **Instantiation in Behavioral Cloning**

Next, we apply SIMILE to GAIL (Ho & Ermon, 2016), a recently proposed baseline of BC. GAIL assumes the expert trajectories are sampled from an implicit policy $\pi_\mathrm{E}$, and trains a discriminator $D(\cdot, \cdot)$ to distinguish whether each state-action pair $(s, a)$ is likely to be generated from $\pi_\mathrm{E}$. The entire learning algorithm is an adversarial training process in which the parameters of the discriminator $D$ and policy $\pi$ are updated alternatively. Applying SIMILE to GAIL is straightforward, for which we only need to replace the original sampling strategy $a_t \sim \pi(s_t, \cdot)$ with $a_t \sim \tilde{\pi}(s_t, \cdot \mid s_0, a_0, \ldots, s_{t-1}, a_{t-1})$. The modified pipeline is illustrated in Algorithm 2.

- **Enhancing Sequential Information in the Classifier**

In both IRL and BC, the classification module ($R$ or $D$) plays an important role in estimating total rewards by both the expert and agent, and adjusting the reward function to lean towards the expert. When SIMILE is added, this total reward estimation can also be incorporated. For this purpose, we add an individual output to each LSTM cell for the reward in this step, and summarize all these rewards into the final discriminator. This is to fuse a sequence of weak classifiers into a strong one, which we use the adaptive boosting (AdaBoost) algorithm (Freund & Schapire, 1997). The point is that, AdaBoost captures the importance of each state (with the help of LSTM) and adjusts the weight on it. Consequently, more important states are better optimized. In experiments, this strategy produces small but consistent accuracy gain, so we will apply it by default in the following experiments.

---

**Algorithm 2** Applying SIMILE to GAIL (Ho & Ermon, 2016)

1: **Input:** expert trajectories $\mathcal{T} = \{\hat{\tau}_1, \hat{\tau}_2, \ldots, \hat{\tau}_K\} \sim \mathcal{P}(\pi_\mathrm{E}, P)$, learning rates $\eta_1$ and $\eta_2$;
2: Initialize: policy $\tilde{\pi}$ parameterized by $\boldsymbol{\theta}_0$, discriminator $D$ parameterized by $\boldsymbol{\omega}_0$;
3: **for** $m = 1, 2, \ldots, M$ **do**
4:    $s_{m,0} \sim P_0(\cdot), T_m \leftarrow 0$;
5:    **while** task is not accomplished **do**
6:      $a_{m,t} \sim \tilde{\pi}(s_{m,t} \mid s_{m,0}, a_{m,0}, \ldots, s_{m,t-1}, a_{m,t-1})$;
7:      $s_{m,t+1} \sim P(\cdot \mid s_{m,t}, a_{m,t}), T_m \leftarrow T_m + 1$;
8:    **end while**
9:    $\tau_m \leftarrow (s_{m,0}, a_{m,0}, \ldots, s_{m,T_m-1}, a_{m,T_m-1}, s_{m,T_m})$;
10:   $\boldsymbol{\omega}_{m+1} \leftarrow \boldsymbol{\omega}_m - \eta_1 \times \{\mathbb{E}_{\tau_m}[\nabla_{\boldsymbol{\omega}_m} \ln D(s,a)] + \mathbb{E}_{\hat{\tau}_k \in \mathcal{T}}[\nabla_{\boldsymbol{\omega}_m} \ln(1 - D(s,a))]\}$;
11:   $\boldsymbol{\theta}_{m+1} \leftarrow \boldsymbol{\theta}_m + \eta_2 \times \{\mathbb{E}_{\tau_m}[\nabla_{\boldsymbol{\theta}_m} \ln \pi(a \mid s) \cdot \mathbb{E}_{\tau_m}[\ln D(s',a') \mid s_0' = s, a_0' = a]]\}$;
12: **end for**
13: **Output:** the learned policy $\tilde{\pi}$ parameterized by $\boldsymbol{\theta}_M$.

---

### 3.4 Discussions and Relationship to Prior Work

It has been proved that both IRL and BC are equivalent to GAN (Finn et al., 2016; Ho & Ermon, 2016), in which $\tilde{\pi}$ is the generator, and $D$ or $R$ is the discriminator, respectively. Obviously, SIM-ILE does not change such property, so that we can still analyze the convergence of these iterative algorithm using GAN. As we shall see in Section 4.3, the sampling strategy of SIMILE enjoys a higher training efficiency, making it possible to achieve reasonable performance with fewer expert demonstrations and/or training iterations.

The fundamental assumption of SIMILE is that a single state variable may not provide sufficient information for making a decision. This issue also exists in RL algorithms (Bakker, 2002), but we argue that RL algorithms are able to alleviate this drawback, because the ground-truth reward function serves as a strong supervision at each state-action pair. In IL, however, the supervision becomes much weaker, with an estimated classifier being used to distinguish the agent from the expert, which largely magnifies the inaccuracy in the sampling stage and, consequently, the noise in training data. This hypothesis is partially verified in Section 4.1 which shows the impact of SIMILE on IRL and BC. Compared to IRL which provides an estimation of total reward, BC trains a discriminative function in each state-action pair and thus the supervision is stronger. Consequently, SIMILE show much larger improvement on IRL than on BC.

On the other hand, Monte-Carlo (MC) is a widely used sampling strategy to estimate the total reward. It assumes that the impact of an action will decay with time, and thus introduces a discounted factor $\gamma$ to attenuate the reward terms obtained in the far future. SIMILE provides complementary information, which considers the impact of previous state-action pairs. A similar idea appeared in SeqGAN (Yu et al., 2017) which used MC search to complement sequences in a discrete action space. SIMILE shares the same goal with MC, which is to improve sampling efficiency, so that the agent can get more valuable information from a limited amount of sampled data.

## 4 Experiments

### 4.1 Setting and Baselines

We evaluate SIMILE in OpenAI Gym (Brockman et al., 2016), a popolar benchmark for RL algorithms. Following GAIL (Ho & Ermon, 2016), we choose a few robot controlling tasks from the classic control and MuJoCo subclasses, both of which are simulated using physics engines (Todorov et al., 2012). Each task is provided with a ground-truth cost function, and we generate a number of expert demonstrations for each task by running the TRPO algorithm (Schulman et al., 2015) on these true costs. We train individual models for each task, and the number of iterations follow the same setting as the baselines.

As described in Section 3.3, there are two baselines for BC and IRL, namely GAIL (Ho & Ermon, 2016) and the vanilla IRL, respectively. We do not use the other baselines FEM and GTAL in the GAIL paper, due to their poor performance compared to GAIL. In all scenarios, we randomly choose $70\%$ data for training and use the remaining $30\%$ for testing (the training/testing split remains unchanged). Following GAIL, the basic policy network is the same for all tasks, using 100 neurons at each hidden layer, and $\tanh(\cdot)$ activation to provide non-linearity. We train all models from scratch (*i.e.*, using random Gaussion noise to initialize all weights), and use the Adam optimizer (Kingma & Ba, 2015) with a mini-batch size of 128.

### 4.2 Quantitative Results

Quantitative results on 8 robot controlling tasks are summarized in Table 1. We first observe the baselines. Vanilla IRL often performs worse compared to vanilla BC and its improved version, GAIL. The difference lies in that IRL recovers the reward function $R$ and calls the RL inner-loop, while BC directly constructs a discriminative function $D$ directly. From another perspective, BC allows joint optimization over two modules which often leads to better performance. GAIL goes one step beyond vanilla BC because it introduces adversarial training which improves stability.

SIMILE brings significant improvement beyond vanilla IRL. In 7 out of 8 tasks, IRL+SIMILE works better than vanilla BC. The only exception, CartPole, is the simplest task overall in which

|  | Acrobot | CartPole | MountainCar | Ant |
|---|---|---|---|---|
| Expert | $-75.25 \pm 10.94$ | $200.00 \pm 0.00$ | $-98.75 \pm 8.71$ | $4228.37 \pm 424.16$ |
| Random | $-200.00 \pm 0.00$ | $18.64 \pm 7.45$ | $-200.00 \pm 0.00$ | $-69.68 \pm 111.10$ |
|  | $K_{\text{Acrobat}} = 10$ | $K_{\text{CartPole}} = 10$ | $K_{\text{MountainCar}} = 10$ | $K_{\text{Ant}} = 25$ |
| IRL | $-120.49 \pm 10.42$ | $78.35 \pm 9.35$ | $-155.70 \pm 9.07$ | $2595.18 \pm 112.61$ |
| +SIMILE | $\mathbf{-79.90} \pm 3.88$ | $\mathbf{165.85} \pm 11.91$ | $\mathbf{-102.87} \pm 6.39$ | $\mathbf{3587.41} \pm 18.26$ |
| Vanilla BC | $-95.09 \pm 33.33$ | $177.19 \pm 52.83$ | $-123.14 \pm 28.26$ | $3235.73 \pm 1186.38$ |
| GAIL | $\mathbf{-78.91} \pm 15.76$ | $\textcolor{red}{\mathbf{200.00}} \pm 0.00$ | $-100.83 \pm 11.40$ | $4132.90 \pm 878.67$ |
| +SIMILE | $-81.61 \pm 4.46$ | $\textcolor{red}{\mathbf{200.00}} \pm 0.00$ | $-99.56 \pm 4.79$ | $\mathbf{4143.47} \pm 23.78$ |

|  | HalfCheetah | Hopper | Humanoid | Walker2D |
|---|---|---|---|---|
| Expert | $4463.46 \pm 105.83$ | $3571.38 \pm 184.20$ | $9575.40 \pm 1750.80$ | $6717.08 \pm 845.62$ |
| Random | $-282.43 \pm 79.53$ | $14.47 \pm 7.96$ | $122.87 \pm 35.11$ | $0.57 \pm 4.59$ |
|  | $K_{\text{HalfCheetah}} = 25$ | $K_{\text{Hopper}} = 25$ | $K_{\text{Humanoid}} = 240$ | $K_{\text{Walker2D}} = 25$ |
| IRL | $2403.62 \pm 141.58$ | $2640.48 \pm 91.16$ | $6391.27 \pm 329.96$ | $4748.97 \pm 165.62$ |
| +SIMILE | $\mathbf{3955.02} \pm 149.19$ | $\mathbf{3483.67} \pm 20.47$ | $\textcolor{red}{\mathbf{10220.98}} \pm 165.77$ | $\mathbf{6329.22} \pm 179.07$ |
| Vanilla BC | $3718.58 \pm 1856.22$ | $3383.96 \pm 657.61$ | $5660.53 \pm 3600.70$ | $1599.36 \pm 1456.59$ |
| GAIL | $\textcolor{red}{\mathbf{4840.07}} \pm 95.36$ | $3560.85 \pm 3.09$ | $\textcolor{red}{\mathbf{10361.94}} \pm 61.28$ | $\textcolor{red}{6832.01} \pm 254.64$ |
| +SIMILE | $\textcolor{red}{4752.63} \pm 84.41$ | $\textcolor{red}{\mathbf{3612.33}} \pm 43.10$ | $9689.83 \pm 153.62$ | $\textcolor{red}{\mathbf{8089.60}} \pm 67.29$ |

Table 1: The performance of different IL approaches measured by negative total costs (larger is better) in eight OpenAI Gym environments. The upper and lower parts summarize the results of different tasks. In each case, we use the standard number of demonstrations in the GAIL paper. Red numbers indicate the performance on or above the human level.

the cart only needs to trace the angle of the pole, so BC fits it well. In 4 out of 8 tasks (Acrobot, MountainCar, Hopper and Humanoid), the performance is even close to GAIL (by "close" we mean the score difference between GAIL and IRL+SIMILE is less than 3% of that between GAIL and random). Note that these four tasks are often more complex (*e.g.*, Acrobot and MountainCar compared to CartPole), so we can claim that SIMILE, by referring to a few prior states, can better complement IRL in understanding complex scenes.

On the other hand, SIMILE also improves the overall performance of GAIL – among 8 tasks, 4 are better, 1 remains the same, and 3 are slightly worse. In Hopper, SIMILE boosts the performance of GAIL to surpass the expert performance. In Walker2D, given that GAIL is already slightly better than the expert, SIMILE further boosts the performance by nearly 20%. Investigating deeper into Walker2D, we find that the aim of this is to make a 2D robot walk as fast as possible, but the robot also needs to maintain its balance while it hops and moves forward (falling onto the ground will lead to game over immediately). In this case, there may exist much hidden information, such as the acceleration of each link, that can not be predicted from a single state variable. Introducing sequential information is a good complementariness in this scenario, which corresponds to the impressive performance gain.

### 4.3 Training Efficiency: Iteration Count and Time Cost

We evaluate the computational costs in training these models. In average, SIMILE reduces the number of training iterations to achieve convergence by about 5%. This is seemingly small, but in potential real-world applications, interaction with the environment is increasingly expensive and time-consuming, and so the advantage fewer training iterations will become more significant.

On the other hand, introducing an LSTM cell to replace the original policy network brings slight extra computation, which in average increases the time costs in each training iteration by around 5%. This approximately cancels out the 5% fewer training iterations, making the overall time cost of SIMILE comparable to the baseline algorithms in these virtual environments. However, in real-world RL applications, these extra costs are almost ignorable as the major overhead lies in agent-environment interactions.

### 4.4 TESTING STABILITY: VARIATION OF PERFORMANCE

In these experiments, another interesting finding is that SIMILE makes the testing performance more stable, *i.e.*, the overall standard deviation is smaller than the baseline. In particular, in Ant, GAIL produces a very large deviation of 878.67, which may be caused by the instability of expert demonstrations (deviation is 424.16), but SIMILE reduces it to merely 23.78. Also, in the best case Walker2D, SIMILE is much more stable than both the baseline and the expert. These advantages suggest the benefit of sequential information, which reduces the ambiguity in the training and testing process – even two situations looks similar according to the state variable, SIMILE can refer to hidden information so as to distinguish them and take different actions.

## 5 CONCLUSIONS AND FUTURE WORK

This paper proposes an approach named Sequential IMItation LEarning (**SIMILE**) to effectively introduce sequential information into conventional IL algorithms. It works by formulating the decision process into a recurrent model, and then using an LSTM to capture both long-term and short-term information. This generalized framework can be instantiated on IRL and BC, two popular variants of IL. The entire algorithm can be optimized in an end-to-end manner. Experiments on a series of robot controlling tasks verify the effectiveness of our approach. In addition to the improved performance, SIMILE enjoys two-fold benefits, namely, faster convergence and more stable testing performance. These advantages demonstrate its potential applications to real-world RL problems.

The success of SIMILE suggests that a good agent (*e.g.*, a human) often takes rich information into consideration before making decisions. However, current RL or IRL algorithms are mostly executed on one single state which is far from sufficient. Our study goes one step further along in this direction, but there remains much more to explore. For example, our recurrent model only memorizes a few history data, yet it may require an individual module to formulate some implicit factors to complement the state variable. Another interesting problem is to investigate the transfer ability of IL systems. For example, if an agent is trained in an environment and then transferred to another one with a different set of physical parameters (*e.g.*, the mass of an object, the speed of enemies, *etc.*), it remains unclear how well it can perform. There were some preliminary work in this field (Fu et al., 2018), and we pave a new direction which is to introduce additional (*e.g.*, sequential) information. These topics are left for future work.

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
