# OpenReview forum: "SIMILE: Introducing Sequential Information towards More Effective Imitation Learning"
_ICLR.cc/2019/Conference_

### Official Review · AnonReviewer1 · 2018-10-30

**Rating:** 4
**Confidence:** 4

**Review:**

The paper proposes to integrate sequential information into imitation learning techniques.  The assumption is that mostly all the IL techniques are learning a policy which depends on state at time t, while the information contained in this state may be not sufficient to choose the right action (actually, this is the POMDP setting, the notion of POMDP not appearing in the paper....). The authors thus propose to use a recurrent neural network to encode the state by aggregating past information, instead of just using the features of the state at time t. They thus instantiate this idea on different methods and show that, on some problems, this approach can increase the quality of the final policy.

Actually, the contribution of the paper is a simple extension of existing methods: using a RNN instead of a simple NN in imitation learning models. First of all, when dealing with classical environments such as Atari, many papers propose to use the last N frames as a state encoding (instead of the last frame), following the same intuition. The studied setting thus corresponds to the PO-MDP case and using a RNN in POMDP is for example what is done in  [Merel etal. 2017]. Moreover, the problem of imitation learning (and particularly inverse RL) in POMDP has been of the interest of many papers like [Choi et al. 2008] for instance and many more, and it is unclear what is the positioning of this paper w.r.t existing works. Since the paper proposes just to encode history with a RNN, the proposed solution lacks of originality, and the contribution of the paper in term of model is quite low.  But the authors explain how this can be instantiated in three different settings (IRL, GAIL and BC) -- note that the section concerning the use of Adaboost is not clear and could be better described -- which can be of the interest of the community.
Concerning the experiments, I don't understand what is the split between training and testing data. Is it pairs of state-action coming from the experts ? or trajectories ? Moreover, I don't understand why these environments correspond to POMDP cases and the authors have to give details on that. For instance, mountain-car is clearly not a POMDP problem in its classical shape, nor Acrobot. As if, it makes the experiments very difficult to reproduce. The interest of using the RNN to encode history does not seem clear for each of the cases since it often degrades the final performance, so I don't know exactly what insights I can extract from the paper.

Pro:
* The approach is proposed for IRL, GAIL and BC

Cons:
* Lack of positionning w.r.t POMDP litterature
* Lack of details in the experiments, and lack of good experimental results
* Low contribution in term of model


[Merel et al. 2017]  Learning human behaviors from motion capture
by adversarial imitation
[Choi et al.] Inverse Reinforcement Learning in Partially Observable
Environments

---

> ### Author Response · Authors · 2018-11-24
> **We thank the reviewer for valuable comments**
>
> We thank the reviewer for valuable comments. While the idea is simple and the contribution in term of model seems small, we posed an important problem that sequential information is important in the RL-related approaches.
>
> We are sorry that we have missed the connection between this work and POMDPs. The mentioned papers will be cited and discussed thoroughly. We provide a simple solution to RL in POMDPs, which is the major contribution of this work.
>
> Regarding "lack of good experimental results", we achieved much better scores in most scenarios, and in some of them, we achieved better performance than human experts.
>
> Regarding experimental details, we can provide more in the revised version.
>
> Thanks again for helping us improve the quality of this paper.

---

### Official Review · AnonReviewer3 · 2018-11-02
**Straightforward extension of learning-from-demonstration approaches to exploit recurrent neural network**

**Rating:** 4
**Confidence:** 5

**Review:**

The paper puts forward the idea of using a recurrent neural network in algorithms for learning from demonstration in order to take into account sequential information. The authors test it in the inverse reinforcement learning setting and the behavioral cloning setting on different control problems.

I feel the basic idea is really straightforward. Although some promising results are obtained in the experimental setting, I believe the contribution may not be sufficient for a publication at ICLR. Moreover, there are some issues in the writing, e.g.,

- classically, as far as I know, RL is not considered to be a metaheuristic, although I understand that someone could make the case for it.

- although there’s not really a consensus on terminology, I think using imitation learning to define the whole class of problems encompassing IRL and behavioral cloning is not the best. Generally, imitation learning is equated to behavioral cloning. I think a better term for this general class is learning from demonstration. For instance, there are some IRL approaches that don’t try to mimic a demonstrated policy, but aim at learning an even better policy.

- the issue described in the paper about the missing sequential information is due to the fact the authors consider POMDPs and not MDPs. This should be made clearer. I think the authors should also cite the following paper:

@article{ChoiKim11,
	Author = {Jaedeug Choi and Kee-Eung Kim},
	Journal = {JMLR},
	Pages = {691--730},
	Title = {Inverse Reinforcement Learning in Partially Observable Environments},
	Volume = {12},
	Year = {2011}}

- the related work has to be reworked. Kuderer et al. (2013) is not about urban route planning, but deals with learning driving style; Mnih et al. (2015) is not about training multi-agent systems, but introduces DQN; Silver et al. (2016) is about go, not chess. Are TRPO or PPO really off-policy or asynchronous?

- the last section of Sec.3.4 sounds strange. It’s not MC that assumes that the impact of an action decays with time. The discount factor comes from the choice of the total discounted reward criterion.

Other comments:

- in abstract: BL -> BC
- notations issues in (2-5)
- l.6-7, Algo 1: t = T_m?
- The text should be checked for typos.

---

> ### Author Response · Authors · 2018-11-24
> **We thank the reviewer for valuable comments**
>
> We thank the reviewer for valuable comments. While our idea is straightforward, it reveals the importance of introducing sequential information into these scenarios. This topic was not clearly studied before.
>
> We totally agree with, and thank the reviewer on connecting this work with POMDPs. The JMLR'11 paper will be cited and discussed.
>
> Regarding the comments on terminologies, we will remove the statement that RL is a metaheuristic (this is not a major statement in this paper), as well as use a better way of organizing imitation learning, behavioral cloning and inverse reinforcement learning (again, this does not harm the contribution of this paper).
>
> Sorry for the carelessness in the Related Work section. We will improve it.
>
> The last paragraph of Section 3.4 was not well written. The core idea is to emphasize that our way of introducing sequential information is essentially different from the strategy of using a discounted factor. Being straightforward, we are considering removing this paragraph which does not harm the contents of this paper.

---

### Official Review · AnonReviewer2 · 2018-11-03
**First review**

**Rating:** 6
**Confidence:** 3

**Review:**

This paper introduces the use of sequential information (state-action pairs) for enhancing imitation learning, and using recurrent networks (LSTM) in that process.  The authors motivate this by pointing out that while the state information, if Markovian, should contain all information necessary for decision making, with incomplete learners redundant information in the sequential state-action information leading to the current state can be helpful, citing some concrete examples.
After describing a number of variants of this idea, in the context of IRL, BC, etc., the authors conduct a systematic empirical evaluation to assess the effectiveness of the proposal, over the baselines, using a number of RL benchmark problems.
The results are favorable and convincingly show that the proposed sequential enhancement can bring significant improvement in terms of attained rewards, convergence speed and stability in many of the tested cases.
One suggestion I have is that it would be interesting to investigate into the question of how the addition of sequential information adds value is related to the validity of Markovian assumption in each of the problem being considered.
It is a good empirical paper demonstrating the practical use of an idea that is simple but reasonable, and in a way that is substantiated using proper cutting edge framework and baselines.

---

> ### Author Response · Authors · 2018-11-24
> **We thank the reviewer for valuable comments**
>
> We thank the reviewer for valuable comments. We are grateful that the reviewer recognized the value of this work: it adds sequential information which improves the overall performance of both IRL and BC algorithms.
>
> In addition, we totally agree that investigating the relationship between how the addition of sequential information adds value and the validity of Markovian assumption. We will try to provide some qualitative and quantitative analysis in the future revisions.

---

### Meta-Review · Area_Chair1 · 2018-12-13
**Paper should discuss and account for partial observability**

**Confidence:** 4
**Recommendation:** Reject

**Metareview:**

This paper explores the use of sequential information to improve imitation learning, essentially using recurrent networks (LSTM) instead of a simple NN in several existing imitation learning models (BC, GAIL, etc.). On the positive side, the empirical results are good, showing improvement in terms of attained rewards, convergence speed and stability. There are however some significant issues with the way the way the approach is motivated and positioned with respect to existsing work. In particular, the issue described in the paper is due to the fact they consider POMDPs (not MDPs): this should have been more clearly explained. There are also issues with the Related Work section. For these reasons, the paper is not quite ready for publication.